# Psychosocial interventions and their effectiveness on quality of life among elderly persons living with HIV in Africa South of the Sahara: Systematic review and meta -analysis protocol

**Marijanatu Abdulai[1,2], David Owiredu[1,3], Isaac Boadu[4], Philip Teg-Nefaah Tabong[5], Bismark Sarfo[1], Harriet Affran Bonful[1], Adolphina Addo-Lartey[1], Kwadwo Owusu Akuffo[6], Anthony Danso-Appiah[1,3]***

1 Department of Epidemiology and Disease Control, School of Public Health, University of Ghana, Legon, Accra, Ghana, 2 National AIDS/STI Control Programme, Public Health Division, Ghana Health Service, Accra, Ghana, 3 Centre for Evidence Synthesis and Policy, School of Public Health, University of Ghana, Legon, Accra, Ghana, 4 Department of Population, Family and Reproductive Health, School of Public Health, University of Ghana, Legon, Accra, Ghana, 5 Department of Social and Behavioural Science, School of Public Health, University of Ghana, Legon, Accra, Ghana, 6 Department of Optometry and Visual Science, College of Science, Kwame Nkrumah University of Science and Technology, Kumasi, Ghana

* adanso-appiah@ug.edu.gh, tdappiah@yahoo.co.uk

## Abstract

### Background

The number of elderly people living with HIV (EPLHIV) has increased significantly as a result of antiretroviral treatment (ART) and this has brought about a variety of psychosocial challenges that have an impact on their quality of life (QoL). Various psychosocial interventions have been tried or implemented in Sub-Saharan Africa (SSA) to improve QoL of EPLHIV. However, there is paucity of data on the types and effectiveness of these interventions. This systematic review, therefore, aims to explore available psychosocial interventions in SSA and their effectiveness in improving the QoL of EPLHIV.

### Methods

We will search PubMed, PsycINFO, LILACS, Cochrane Library, Google Scholar, HINARI, Africa Journals Online, Scopus and Web of Science to retrieve publications on psychosocial interventions implemented to improve QoL of EPLHIV from inception of the identified databases to 31st December 2023 without language restrictions. Also, supplementary sources such as conference proceedings, preprint repositories, databases of dissertations, as well as WHO and governmental databases can be explored for additional studies. For unpublished studies, trial registries and experts would be contacted, and reference lists of retrieved papers will be manually searched. Retrieved studies will be deduplicated using Mendeley and exported to Rayyan. At least two reviewers will independently select studies, extract data and assess the quality of the included studies using validated tools.

**Data Availability Statement:** No datasets were generated or analysed during the current study. All

relevant data from this study will be made available upon study completion.

**Funding:** The author(s) received no specific funding for this work.

**Competing interests:** The authors have declared that no competing interests exist.

**Abbreviations:** ART, Antiretroviral treatment; ARVs, Antiretrovirals; CBT, Cognitive Behavioural Therapy; CDC, Centers for Disease Control and Prevention; CI, Confidence Interval; EPLHIV, Elderly persons living with HIV; HIV, Human Immunodeficiency Virus; HRQOL, Health-Related Quality of Life; LMICs, Low- and Middle-Income Countries; MD, Mean Difference; OoL, Quality of Life; OR, Odds Ratio; PICOS, Patient/Population. Intervention. Comparison, Outcomes; PLHIV, Persons living with HIV; PRISMA, Preferred Reporting Items for Systematic Review and Meta-Analysis; PROSPERO, Prospective Register for Systematic Reviews; QoLS, Quality of Life Scale; RCT, Randomized Controlled Trials; RR, Risk Ratio; SDGs, Sustainable Development Goals; SSA, Sub-Saharan Africa; UGCESP, University of Ghana Centre for Evidence Synthesis and Policy; WHO, World Health Organisation.

Dichotomous outcomes data will be assessed and reported as odds ratio (OR) or risk ratio (RR) and for continuous outcomes, mean difference (MD) will be used; all reported with their 95% confidence interval (CI). Heterogeneity will be explored graphically by inspecting the overlapping of CIs and assessed quantitatively using the $I^2$ statistic.

## Expected outcomes

This systematic review will be the first to rigorously identify psychosocial intervention on QoL of EPLHIV in SSA and assess their effectiveness with the aim to provide regional and country- specific data that will inform the selection and implementation of appropriate and socially acceptable policies across countries in SSA. Key findings of the review are expected to contribute critical evidence on availability, types and effectiveness of psychosocial interventions for improving quality of life of vulnerable elderly persons in SSA living with HIV. Furthermore, the review will explore any variation and possible correlates of psychosocial interventions by age, sex, CD4 count (if available), setting and geographic location within SSA that will provide healthcare professionals with reliable evidence, with the ultimate goal of inspiring countries in SSA to adopt innovative interventions to improve HIV care.

## Trial registration

**Systematic review registration:** The systematic review protocol has been registered in the International Prospective Register for Systematic Reviews (PROSPERO), with registration ID **CRD42021278218**.

## Background

Human Immunodeficiency Virus (HIV) remain global public health challenge claiming many lives despite significant effort to reduce the spread of the infection [1, 2]. Over 38.4 million people were living with HIV in 2021, with 1.5 million new infections and 650 000 HIV- related deaths globally, of which 36.7 million were adults (15 years and above), with countries in Africa South of the Sahara (SSA) accounting for 25.4 million deaths (66.1%) [3, 4].

The introduction of antiretroviral therapy (ART) has resulted in a significant reduction in HIV-related morbidities and mortalities whilst increasing life expectancy among people living with HIV (PLHIV) [5]. Antiretrovirals are highly effective in suppressing the replication of HIV, reducing the viral load in the body. This helps slow the progression of HIV disease, preserving immune function and preventing further damage to the immune system [6]. Empirical evidence suggests that the number of PLHIV on lifelong ART has tripled from 7.5 million in 2010 to 27.5 million in 2020, of which a higher proportion (61%) live in East and Southern Africa [7, 8]. Despite the benefits of ART, HIV infection and related psychosocial effects still impact the quality of life (QOL) of PLHIV significantly, especially the elderly who are already overwhelmed with the challenges of ageing and associated comorbidities [9, 10].

Psychosocial well-being, a multidimensional concept comprising psychological, social, and other subjective mental states, remains a big challenge for PLHIV generally, but critically among EPLHIV [11]. Globally, the prevalence of depression, anxiety and mood disorders accounts for about 33% among PLHIV [12, 13]. A systematic review of PLHIV in sub-Saharan Africa found pooled estimates of anxiety and depression to be 32% and 9%, respectively [14, 15]. Psychological symptoms usually lead to nonadherence to ART, poor rating of QoL and

increased HIV-related deaths [16]. In North America and Europe, evidence showed that a positive HIV test result alone and the perception of life changes associated with a life-threatening, chronic illness led to episodes of depression and anxiety [13, 17, 18].

It was not surprising that the 2021 Technical report of the WHO emphasized the increased risk of mental health problems among PLHIV, particularly EPLHIV, which usually leads to less motivation to adhere to treatment, increased risk behaviours, and decreased engagement with HIV prevention measures [19, 20]. Therefore, a top priority of the 2021–2026 Global AIDS Strategy has been the integration of mental health and psychosocial support with HIV services, especially those led by communities to help improve the QoL of PLHIV [20, 21].

People living with HIV are more susceptible to HIV-related neurocognitive dysfunction due to mitochondrial damage, similar to a common occurrence in elderly persons [22]. They may be accompanied by undesirable physical and psychological changes affecting QoL [23, 24]. Elderly people living with HIV are particularly vulnerable because their age is a vital precursor of adverse mitochondria outcomes. Psychosocial interventions such as cognitive behavioural therapy (CBT), supportive interventions, meditation and stress management to improve QoL have been developed for EPLHIV in developed countries with some positive outcomes [25–28]. Despite the evidence suggesting the need to focus on improving the QoL of EPLHIV because of their increased vulnerability and needs, there is paucity of data in SSA about the availability and effectiveness of psychosocial interventions for improving the QoL of this unique group [29–32]. Undoubtedly, psychosocial interventions targeted to EPLHIV are critical to achieving SDG 3 [33], however, this is simply a neglected issue and not on the policy agenda of most countries across SSA [33, 34].

Although several systematic reviews on the impact of psychosocial intervention on QoL among the elderly exist but these have focused on chronic diseases such as cancers [32, 35], chronic heart failure and type II diabetes mellitus [22, 36, 37] but there is little or no evidence on psychosocial interventions targeting the QoL of EPLHIV in Africa South of the Sahara. Given the high number of EPLHIV in Africa South of the Sahara and the trend pointing to a further increase, it is of utmost importance to identify and assess, through systematically synthesized data, available psychosocial interventions and their effectiveness in Africa South of the Sahara to inform the sound and context-specific policies for improving QoL of EPLHIV [13, 20, 36].

This systematic review aims to document available psychosocial interventions to improve the QoL of EPLHIV in SSA and assess their effectiveness. In particular, the review will answer the following specific questions; what psychosocial interventions on QoL are available in SSA? 2) which of them are effective in improving QoL of EPLHIV? and 3) what are the correlates of QoL EPLHIV in the Sub-Saharan African context?

## Materials and methods

This protocol has been prepared in line with standard systematic review manuals and reported per the reporting guidance provided in the Preferred Reporting Items for Systematic Reviews and Meta-Analyses Protocol (PRISMA-P) statement [38] (S1 Table). The full review will be reported per the PRISMA guidelines [39] and the PRISMA flow (S1 Fig).

### Patient and public involvement

The review questions and outcome measures have been developed collaboratively with the relevant patients and public involvement. They are informed by their priorities, experience, and preferences per the Guidance for Reporting Involvement of Patients and the Public (GRIPP2)

checklist. The review findings will be shared with relevant wider patient communities, who will also be involved in disseminating the results.

## Criteria for including studies for inclusion in this systematic review

**Type of studies.** Randomized Controlled Trials (RCTs), quasi-experimental studies, and observational studies, including cohort, case-control, and cross-sectional studies that reported on psychosocial interventions and quality of life of elderly persons living with HIV in Africa South of the Sahara, will be eligible for inclusion. If sufficient RCTs are obtained, the review will be restricted to only RCTs. Reviews, opinions, commentaries, case reports and case series will not be eligible for inclusion in this review. Whilst reviews are not eligible for inclusion, we will review them to identify potentially relevant primary studies missed by our searches. For a multi-country study, we will extract the studies conducted in countries in Africa South of the Sahara for inclusion in this review. If the study reported a country or regional-level estimate, it would not be eligible for inclusion as such studies do not have a well-defined source population. In cases where the results of a multi-country study have been lumped together and there is no way of disaggregating the data, such studies will not be included.

**Participants.** The population for this systematic review is elderly persons aged 50 years and older receiving ART for HIV and living in SSA. The Centre for Disease Control CDC has defined elderly PLHIV as those aged 50+ years and not 65+ as classified for the general population by the WHO. This CDC classification for older HIV patients is that PLHIV age faster than the non-HIV population [40, 41]. The HIV status of participants to be included should have been determined by prequalified testing approach, an algorithm set out by the WHO and adopted by member countries [42]. Participants should have been registered at a recognized health facility providing antiretroviral treatment services and receiving their routine ARVs. HIV-positive clients who are not on ARVs, are pregnant or breastfeeding, will be excluded. No restrictions will be placed on participants' gender, ethnicity, or other demographic characteristics.

**Intervention.** All psychosocial interventions to improve QoL among EPLHIV will be eligible for inclusion. Interventions will include, but are not limited to, the following: 1) social support, 2) psychotherapy, 3) psychoeducation and counselling, and 4) cognitive behavioural therapy [43, 44].

Psychosocial interventions for QoL improvement administered individually, in groups, or a combination of the two through mobile technology, online platforms, healthcare professionals, and social workers [45] will be eligible for inclusion in the review. Psychosocial interventions not specifically targeting participants' QoL will be excluded.

**Comparator.** The comparator will be elderly people aged 50 years and above living with HIV in Africa South of the Sahara receiving ARVs (or standard treatment for HIV with no additional psychosocial interventions targeted to improving their QoL.

**Outcomes.** *Primary outcomes.* The primary outcome of this review is "improvement in quality of life". There are about 150 instruments available for measuring QoL [46]. However, this review will assess the five main QoL measurement scales, namely, the Quality of Life Scale (QoLS) by Flanagan 1978 [47], the McGill Quality of Life Questionnaire was developed by Cohen et al in 1996 and expanded in 2019 [48], the Health-Related Quality of life questionnaire developed by the CDC in 2000, the World Health Organization Quality of Life Instrument [49] and the Global Quality of Life Scale developed in 1996 [50]. These measurements have been selected because of their psychometric properties and wide application across geographic locations and contexts [51].

Secondary outcomes.

- Availability of tools in SSA

- Correlates of psychosocial interventions on QoL

**Adverse events.** All adverse events reported by the included studies, especially from the RCTs, will be summarized in this systematic review.

### Searches for the identification of studies

**Electronic searches.** We will search all relevant electronic databases for relevant primary studies on psychosocial interventions to improve the quality of life of elderly persons living with HIV. The primary databases to be searched are PubMed, LILACS, CINAHL and Cochrane Library. The searches will be done from the inception of the databases through 31st December 2023, without language restrictions. Other sources such as Google Scholar, African Journals Online, JSTOR, SCOPUS, HINARI, and dissertation and preprint databases will also be searched. The key search terms and concepts to be used are psychosocial intervention, quality of life, HIV, elderly people living with HIV, EPLHIV, antiretroviral therapy and sub-Saharan Africa used with all applicable synonyms and alternative terms. All 48 sub-Saharan African countries based on the World Bank classification will be included individually as search terms in the search (see Table 1 for the search strategy developed for PubMed, which will be adapted for the other databases).

**Other sources to be searched.** Reference lists of relevant studies and published systematic reviews on the subject will be manually searched for potentially eligible studies. Additionally, experts in the field will be contacted for completed but unpublished studies.

**Managing the search output and selecting studies.** All results from the various searches will be collated and deduplicated using the Mendeley reference manager (https://www.mendeley.com). The results will then be exported to Rayyan QCRI [52], a systematic review web app: for screening and selection of studies. The titles and abstracts of articles will be screened independently by at least two review team members using the study selection flow

**Table 1. Search strategy developed for PubMed (to be adapted for the other databases).**

| Search | Query | Results |
|---|---|---|
| | Search: (#7) AND (#8) | |
| | Search: (((((((((((((((((((((((((((((((((((((((("sub-Saharan Africa") OR (SSA)) OR (Angola)) OR (Benin)) OR (Botswana)) OR ("Burkina Faso")) OR (Burundi)) OR (Cameroon)) OR ("Cape Verde")) OR ("Central African Republic")) OR (Chad)) OR (Comoros)) OR (Congo)) OR ("Cote d'Ivoire")) OR (Djibouti)) OR ("Equatorial Guinea")) OR (Ethiopia)) OR (Gabon)) OR ("The Gambia")) OR (Ghana)) OR (Guinea)) OR ("Guinea-Bissau")) OR (Kenya)) OR (Lesotho)) OR (Liberia)) OR (Madagascar)) OR (Malawi)) OR (Mali)) OR (Mauritania)) OR (Mauritius)) OR (Mozambique)) OR (Namibia)) OR (Niger)) OR (Nigeria)) OR (Rwanda)) OR ("Sao Tome and Principe")) OR (Senegal)) OR (Seychelles)) OR ("Sierra Leone")) OR (Somalia)) OR ("South Africa")) OR (Sudan)) OR (Swaziland)) OR (Tanzania)) OR (Togo)) OR (Uganda)) OR (Zaire)) OR (Zambia)) OR (Zimbabwe) | |
| | Search: (#5) AND (#6) | |
| | Search: (((((((adults[Title/Abstract]) OR (elderly[Title/Abstract])) OR (aged[Title/Abstract])) OR ("older people"[Title/Abstract])) OR ("matured people"[Title/Abstract])) OR ("matured persons"[Title/Abstract])) OR ("grown-ups"[Title/Abstract])) OR ("older persons"[Title/Abstract]) | |
| | Search: (#3) AND (#4) | |
| | Search: (((((("HIV/AIDS"[Title/Abstract]) OR ("Human immunodeficiency virus"[Title/Abstract])) OR ("Acquired immune deficiency syndrome"[Title/Abstract])) OR (HIV[Title/Abstract])) OR (AIDS[Title/Abstract]) | |
| | Search: (#1) AND (#2) | |
| | Search: (((((((("quality of life"[Title/Abstract]) OR (QOL[Title/Abstract])) OR (wellbeing[Title/Abstract])) OR (well-being[Title/Abstract])) OR (welfare[Title/Abstract])) OR (happiness[Title/Abstract])) OR ("standard of living"[Title/Abstract])) OR ("living standard"[Title/Abstract])) OR (satisfaction[Title/Abstract]) | |
| | Search: (((((Psychosocial Interventions[Title/Abstract]) OR (programs[Title/Abstract])) OR (programmes[Title/Abstract])) OR (strategy[Title/Abstract])) OR (strategies[Title/Abstract])) OR (innovations[Title/Abstract]) | |

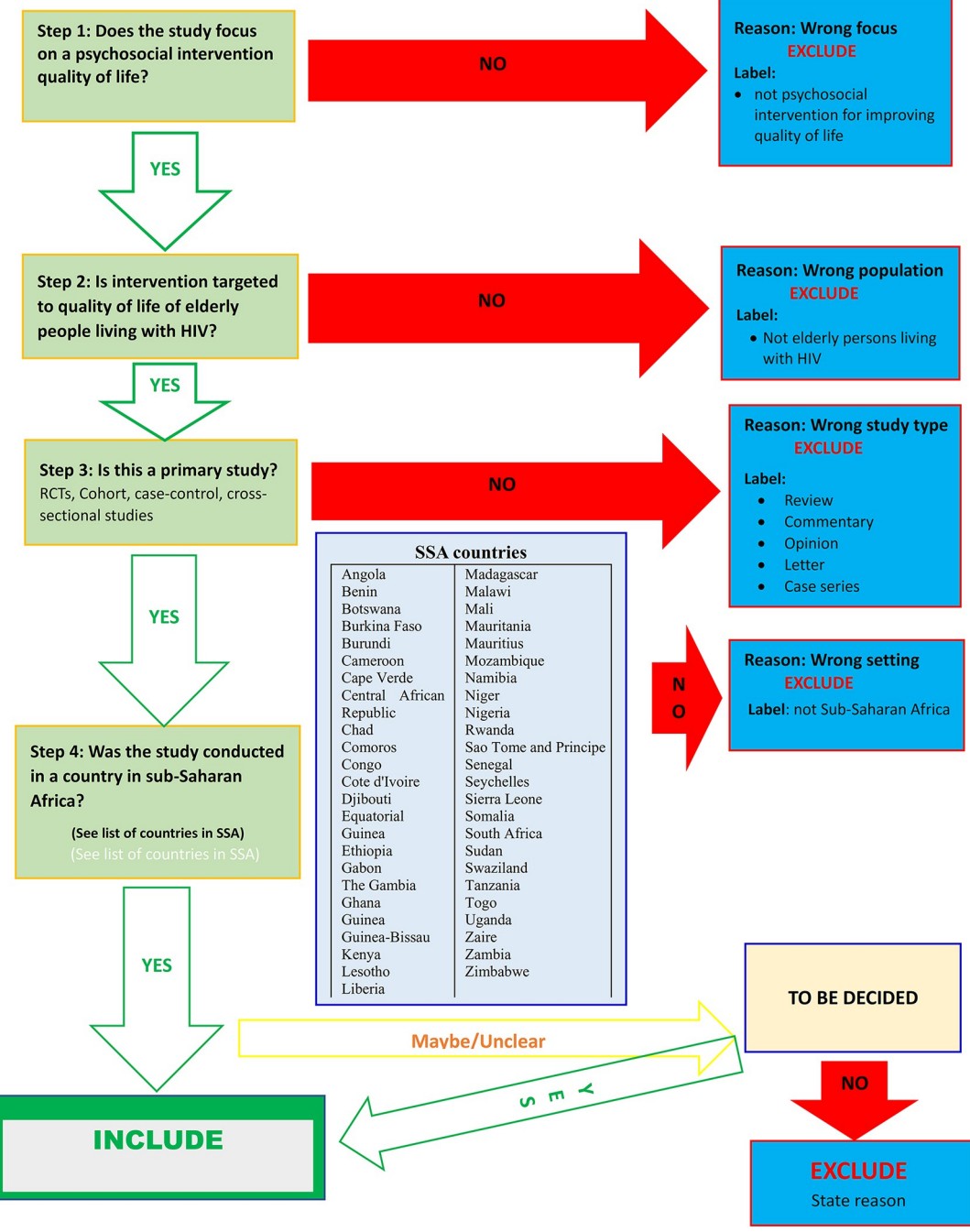

**Fig 1.**

chart developed from the inclusion/exclusion criteria (Fig 1). Where a full-text article cannot be retrieved or is not accessible from the online databases, the team will contact the primary authors. If they fail to help, we will seek assistance from the University of Ghana Library through their inter-library policy. A PRISMA flow chart [53] will be provided to show details of the study flow through the study selection process. Any disagreements will be resolved through discussion between the reviewers.

**Data extraction and management.** Data will be extracted using a pretested form developed in Microsoft Excel [54]. The following study characteristics will be extracted: title, author, year of publication, study area/setting, the country where the study was conducted, study design, sample size, age of participants, and ART status. Then, pertinent data on the interventions and outcomes, such as the type of psychosocial intervention and scale used in measuring QoL, will be collected. Data extracted during the piloting stage will be checked and compared between the data extractors for accuracy and consistency before finally applying the form to the rest of the studies. In the case of more than two reviewers doing the data extraction, 10% of the included studies will be sampled and extracted independently by an experienced reviewer in the team as a quality check. Where necessary, the data extraction form will be updated based on experience learned and all changes made will be documented transparently. Any disagreements between reviewers will be resolved through discussion.

**Assessment of quality of the included studies.** The Cochrane risk of bias tool available in Cochrane Handbook and Review Manager V5.4 will be used to assess the risk of bias in each of the included RCTs in six domains: sequence generation, allocation concealment, blinding (investigators, outcome assessors, and participants), incomplete outcome data, selective outcome reporting and other sources of bias. For each domain, we will make a judgment of 'low risk of bias, 'high risk of bias or 'unclear risk of bias. All the risk of bias domains will be considered to assign each study a low, high, or unclear risk of bias. The Cochrane tool will be used in addition to the quality assessment tool for prevalence studies developed by Hoy et al. (2012) [55] (S2 Table) that assesses the risk of bias in observational studies on four domains: selection bias, non-response bias, measurement bias, and bias related to data analysis. This will be done independently by at least two review team members. For each domain, a judgment of 'low risk of bias,' high risk of bias or unclear risk of bias will be made. The overall quality of evidence contributed by each study will be classified as 'low risk of bias,', high risk of bias or unclear risk of bias. Any discrepancy will be resolved through discussion between the reviewers.

**Plan for data synthesis.** The findings from the included studies will be narratively summarized by the PICOS elements, including the type and characteristics of the participants/population, type of studies, type of psychosocial interventions, and outcomes of interest in the characteristics of the included studies. The quantitative data will be analyzed using Stata version 15.0 (Stata Corp LLC, Texas, USA) and Review Manager v5.4. Dichotomous data will be presented as odds ratio (OR) or risk ratio (RR) and for continuous outcome data, mean difference (MD) will be used with all expressed with their respective confidence intervals (CIs). The following process will be used: firstly, standard deviations for the study-specific estimates, where applicable, will be determined from point estimates and the appropriate denominators assuming a binomial distribution. Then the magnitude of heterogeneity between included studies will be assessed quantitatively using the index of heterogeneity ($I^2$ statistics). The $I^2$ values of 25%, 50%, and 75% are considered to represent low, moderate and significant heterogeneity, respectively [56]. The significance of heterogeneity will be determined by the p-value of the $I^2$ statistics, and a p-value of <0.05 will be considered evidence of heterogeneity. For studies with moderate to significant heterogeneity, the random effects model will be used to obtain a pooled estimate of the outcome, and if heterogeneity is low, a fixed-effect model will be used.

**Heterogeneity and subgroup analysis.** Where the heterogeneity detected is significantly high, a subgroup analysis will be performed to detect the possible sources mainly around the characteristics of the PICOS elements (P-participants/population, I-intervention, C-control/comparator, O-outcomes and S-study type). Specially, we will assess heterogeneity around potential confounding variables such as age, study setting (hospital or community-based), sex, and geographical location (Central, East, Southern and West Africa). Sub-group analysis for RCTs will be done carefully in order not to break the randomization code of the trials.

**Handling missing and incomplete data.** We will attempt to extract sufficient data from all included studies. Where data on some pertinent variables are missing, the original authors of the primary studies will be contacted to see if they can provide the missing data. Where original authors are unable to provide the requested information or cannot be reached, we will not compute. Instead, the potential impact of missing data on the findings of the review will be addressed in the discussion section.

## Sensitivity analysis

Sensitivity analysis, the process of testing the robustness of the results of a systematic review [57], will be used to re-analyze the data to determine if the results are sensitive to specific review elements [58, 59]. The domains to be considered in the sensitive analysis include the quality of the included studies, sample size and the meta-analysis technique applied.

## Grading level of evidence

The Grading of Recommendations, Assessment, Development, and Evaluations (GRADE) approach will be used to assess the overall quality of evidence generated from the review for each outcome. The pooled data of each study will be evaluated against the five GRADE considerations; risk of bias, imprecision, inconsistency, indirectness, and publication bias and will be graded as high, high, moderate, low and very low levels of evidence accordingly [60, 61]. High-quality evidence is interpreted as further research is very unlikely to change the confidence in the estimate of effect, while a grading of very low quality implies an estimate of effect is very doubtful.

## Ethics and dissemination

Ethics approval is not required for this review as the work will be carried out on published and unpublished completed primary studies. The findings will be disseminated in a related peer-review journal and presented at conferences. Findings from the systematic review will also serve as a guide for policymakers in decision-making across countries in SSA.

## Discussion

The introduction of ARTs means that people living with HIV will increasingly live longer and face age-related challenges and HIV-related psychosocial problems, including quality of life. This review is the first to attempt to rigorously summarize evidence on psychosocial interventions for QoL of EPLHIV in SSA to provide country and regional- specific estimates of the effect of psychosocial interventions for improving QoL of EPLHIV that will inform the implementation of context and culturally sensitive policies. The review uses rigorous approaches to address issues relating to external validity (generalizability) using the PICOS to succinctly define the characteristics of the participants, interventions, outcomes and types of studies. It uses transparent and robust methods (as specified in the methods) to generate reliable evidence from a very comprehensive search strategy that will attempt to retrieve all potentially relevant studies from a wide range of study designs such as randomized controlled trials (RCTs), cohort, case-control and cross-sectional studies to gather evidence. Subgroup analysis will be performed to explore whether psychosocial interventions vary according to age, sex, setting and geographic location across SSA. It will also explore socio-demographic characteristics, ART histories and interventions sought among EPLHIV. It is anticipated that these review findings will provide healthcare professionals with evidence-based information regarding the effectiveness of psychosocial interventions in enhancing the QoL of EPLHIV. Moreover, the

review has the potential to inspire countries in SSA to adopt innovative interventions to improve HIV care based on reliable evidence.

## Study limitation

One significant limitation is the lack and quality of primary studies. To address this, we will conduct an extensive search across multiple databases, including local and regional sources, to maximize the identification of relevant studies. Additionally, we will employ established tools to assess the quality of the included studies and conduct sensitivity analyses to gauge the potential impact of study quality on our overall results.

Another challenge lies in the diversity of psychosocial interventions, making direct comparisons difficult. To overcome this, we will categorize the types of interventions used in primary studies carefully. Where feasible, we will perform subgroup analyses to explore the effectiveness of different intervention types separately, enabling us to draw more precise conclusions.

Publication bias, where positive results are more likely to be published, is also a concern. To address this, we will actively search for unpublished studies and explore gray literature sources to ensure a comprehensive representation of the evidence.

## Implications of the anticipated study findings

The review team anticipates that the findings from this proposed systematic review will help improve psychosocial interventions to promote the quality of life for elderly persons living with HIV. The findings will also reveal research gaps to inform future research to provide evidence-based solutions to improve health care access. The finding of this proposed study may also inspire countries within the sub region to adopt such innovations to help improve HIV care especially within this current pandemic and impending pandemics.

## Supporting information

**S1 Fig. PRISMA-P 2020 flow diagram.** The PRISMA Flow diagram shows the flow of studies from retrieval from electronic databases and other sources to selection of studies for inclusion in the systematic review.
(TIF)

**S1 Table. PRISMA-P (Preferred Reporting Items for Systematic Review and Meta-Analysis Protocols) 2015 checklist: Recommended items to address in a systematic review protocol.**
(PDF)

**S2 Table. Hoy risk of bias tool.**
(PDF)

## Acknowledgments

This systematic review was prepared as part of the capacity-building initiative of the Centre for Evidence Synthesis and Policy, University of Ghana and Africa Communities of Evidence Synthesis and Translation (ACEST) that jointly train in evidence synthesis and translation across countries in Africa and Low and Middle Income Countries (LMICs). MA was mentored by ADA.

## Author Contributions

**Conceptualization:** Marijanatu Abdulai, Anthony Danso-Appiah.

**Investigation:** Isaac Boadu.

**Methodology:** Marijanatu Abdulai, David Owiredu, Isaac Boadu, Anthony Danso-Appiah.

**Project administration:** Marijanatu Abdulai, Isaac Boadu, Philip Teg-Nefaah Tabong, Bismark Sarfo, Harriet Affran Bonful, Adolphina Addo- Lartey, Kwadwo Owusu Akuffo, Anthony Danso-Appiah.

**Resources:** David Owiredu, Isaac Boadu, Philip Teg-Nefaah Tabong, Bismark Sarfo, Harriet Affran Bonful, Adolphina Addo- Lartey, Kwadwo Owusu Akuffo, Anthony Danso-Appiah.

**Supervision:** Anthony Danso-Appiah.

**Validation:** Marijanatu Abdulai, David Owiredu, Isaac Boadu, Adolphina Addo- Lartey, Anthony Danso-Appiah.

**Writing – original draft:** Marijanatu Abdulai, David Owiredu, Philip Teg-Nefaah Tabong, Bismark Sarfo, Harriet Affran Bonful, Adolphina Addo- Lartey, Kwadwo Owusu Akuffo, Anthony Danso-Appiah.

**Writing – review & editing:** Marijanatu Abdulai, David Owiredu, Isaac Boadu, Philip Teg-Nefaah Tabong, Bismark Sarfo, Harriet Affran Bonful, Adolphina Addo- Lartey, Kwadwo Owusu Akuffo, Anthony Danso-Appiah.

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
