## [Editor Report · Decision Letter 0]

23 Aug 2023

PONE-D-23-19414Effects of Psychosocial Interventions on Quality of Life Among Elderly Persons Living with HIV and AIDS in Sub-Saharan Africa: Systematic Review Protocol

PLOS ONE

Dear Dr. Danso-Appiah,

Thank you for submitting your manuscript to PLOS ONE. After careful consideration, we feel that it has merit but does not fully meet PLOS ONE’s publication criteria as it currently stands. Therefore, we invite you to submit a revised version of the manuscript that addresses the points raised during the review process.

We look forward to receiving your revised manuscript.

Kind regards,

Felix Bongomin, MB ChB, MSc, MMed, FECMM

Academic Editor

PLOS ONE

2. Please amend either the title on the online submission form (via Edit Submission) or the title in the manuscript so that they are identical.

Additional Editor Comments:

Please find attached manuscript with my comments

---

## [Author Response · Author response to Decision Letter 0]

30 Aug 2023

PONE-D-23-19414

Effects of Psychosocial Interventions on Quality of Life Among Elderly Persons Living with HIV and AIDS in Sub-Saharan Africa: Systematic Review Protocol

Dear Dr. Felix Bongomin

Thank you for your prompt attention and the opportunity to submit a revised version of our manuscript entitled "Effects of Psychosocial Interventions on Quality of Life Among Elderly Persons Living with HIV and AIDS in Sub-Saharan Africa: Systematic Review Protocol" for consideration for publication in PLOS ONE. We have addressed all comments point-by-point in our response below. The editor’s comments are presented in normal font, and we respond directly to them in bold black font. All changes to the manuscript are highlighted in red font. 

Yours sincerely,

Danso-Appiah et al.

We have addressed all the points raised during the review process in this revised version of the manuscript and used the opportunity to also do minor corrections.

Thank you, we have followed PLOS ONE style requirements, including file naming.

2. Please amend either the title on the online submission form (via Edit Submission) or the title in the manuscript so that they are identical.

Thank you, we have made the necessary changes and the title on the submission form and that in the manuscript are the same. 

Thank you, we have moved the ethics statement from after “List of abbreviations” (page 19, last paragraph) to the “Methods” (page 17, first paragraph).

We have reviewed the references. Microsoft Excel as suggested by the reviewer has been cited. Also, I replaced the weblink to Rayyan (cited in-text) with the correct bibliographic citation. The total references now stand at 61.

Additional Editor comments

Please find attached manuscript with my comments

We have revised Figure 1 accordingly in our manuscript. 

Reviewer’s comments

1. Remove AIDS all through the manuscript.

We have removed AIDS all through the manuscript as suggested.

2. Use the current recommneded /acceptable terminology to replace SSA.

We have used the current recommended teminology as suggested - Africa South of the Sahara.

3. a systematic review and meta-analysis protocol.

We have revised the title of the protocol accordingly. 

4. Why this start date?

Searches will be conducted to retrieve relevant studies without time (date of publication) limits. This has now been stated in the manuscript.

5. Consider up to 31st December 2023.

Thank you, our final search date has now been changed from “30th September 2023” to “31st December 2023”.

6. Human Immune deficiency virus (HIV) and Acquired Immune Deficiency Syndrome (AIDS) Check how HIV is written in full. 

Thank you, the authors have acknowledged this typographical error and have corrected the full meaning of HIV as Human Immunodeficiency Virus.

7. Delete “AIDS” all through the text.

Thank you for this suggestion, the term AIDS has been deleted from the whole manuscript except for the name of one of the affiliations which cannot be changed.

8. The rationale for this systematic review

Thank you, we have deleted the heading “Rationale for this systematic review”.

9. Microsoft Excel

We have provided citation to “Microsoft Excel. 

10. SUGGEST POSSIBLE LIMITATIOSN AND HOW YOU WILL OVERCOME THEM

Thank you. We have included in the manuscript the anticipated study limitations and the strategies we will adopt in addressing them.

11. SUGGEST IMPLICATIONS OF THE ANTICIPATED STUDY FINDINGS

Thank you. We have included a section on the implication of the review findings in the manuscript.

---

## [Editor Report · Decision Letter 1]

5 Sep 2023

Psychosocial Interventions and their Effectiveness on Quality of Life Among Elderly Persons Living with HIV in Africa South of the Sahara:  Systematic Review and Meta-analysis Protocol

PONE-D-23-19414R1

Dear Dr. Danso-Appiah,

We’re pleased to inform you that your manuscript has been judged scientifically suitable for publication and will be formally accepted for publication once it meets all outstanding technical requirements.

Kind regards,

Felix Bongomin, MB ChB, MSc, MMed, FECMM

Academic Editor

PLOS ONE
---

## [Editor Report · Acceptance letter]

11 Sep 2023

PONE-D-23-19414R1 

 Psychosocial Interventions and their Effectiveness on Quality of Life Among Elderly Persons Living with HIV in Africa South of the Sahara:  Systematic Review and Meta-analysis Protocol 

Dear Dr. Danso-Appiah:

I'm pleased to inform you that your manuscript has been deemed suitable for publication in PLOS ONE. Congratulations! Your manuscript is now with our production department. 

Kind regards, 

on behalf of

Dr. Felix Bongomin 

Academic Editor

PLOS ONE